# Ready-to-Use Supplementary-Food Biscuit Production with Low-Cost Ingredients for Malnourished Children in Sub-Saharan Africa

**DOI:** 10.3390/foods13111614

**Published:** 2024-05-22

**Authors:** Davide Reggi, Gaia Alessio, Andrè Ndereyimana, Andrea Minuti, Giorgia Spigno, Giuseppe Bertoni

**Affiliations:** 1Department of Animal, Nutrition and Food Science (DiANA), Università Cattolica del Sacro Cuore, Via Emilia Parmense 84, 29122 Piacenza, Italy; gaia_95pv@hotmail.it (G.A.); ndereya79@gmail.com (A.N.); andrea.minuti@unicatt.it (A.M.); giuseppe.bertoni@unicatt.it (G.B.); 2Department for Sustainable Food Process (DiSTAS), Università Cattolica del Sacro Cuore, Via Emilia Parmense 84, 29122 Piacenza, Italy; giorgia.spigno@unicatt.it

**Keywords:** child malnutrition, RUSF, protein biscuit, food security, low-income countries, low-cost integration

## Abstract

In Africa, the number of children under 5 years old who suffer from stunting and wasting are, respectively, 61.4 and 12.1 million, and to manage situations like these, emergency food products like RUTF and RUSF (ready-to-use therapeutic/supplementary food) are very useful. The aim of this study was to develop an RUSF biscuit using the low-cost food resources usually present in Sub-Saharan Africa (Burundi and the DRCongo in our case study); we conducted chemical characterization, nutritional evaluation, and a stability trial simulating the usual storage conditions in a rural context to demonstrate that RUSF can be functional also using low-cost ingredients and a simple method of production. The obtained recipes showed good potential in supplying protein integration—17.81% (BUR) and 16.77% (CON) (% as food) were the protein contents—and the protein digestibility values were very high (BUR: 91.72%; CON: 92.01%). Moreover, 30% of the daily requirement was achieved with less than 50 g of both recipes in all the considered ages. Finally, a good shelf-life was demonstrated during the 35-day testing period at 30 °C, considering moisture, texture, and lipid oxidation evolution. Recipes like these, with appropriate changes, could be very useful in all contexts where child malnutrition is a serious problem.

## 1. Introduction

Malnutrition is one of the biggest problems in low-income countries. A total of 45% of the 5.9 million deaths of children under 5 years worldwide can be linked to undernutrition, which is often connected in turn to other causes like infections and parasitosis [1]. The risk of death for a malnourished child is 11 times higher than for a well-nourished child [2]. It is known that malnutrition compromises not only physical development, but also mental/brain development, which is fundamental for the integral development of any human being [3]. For this reason, Black et al. [4] consider the problem of child malnutrition in developing countries not just related to mortality, but particularly to the negative influence on their development, especially regarding the brain; therefore, malnutrition in the first few years of life affects cognitive development, as well as motor and social–emotional development [5]. Other effects of undernutrition, in the longer term, include mental illness, hypertension, diabetes, and impaired working capacity, with consequences for individual health and the standard of living of the affected individuals throughout their life [6,7].

Malnutrition is often linked to household food insecurity (HFI) [8], which is defined as the lack of physical, social, and economic access to sufficient, safe, and nutritious food to meet the needs and food preferences for an active and healthy life [9].

The problem of HFI, despite some improvements, continues to be very worrying, overall, in Africa. In fact, the FAO declared that three-quarters of the population in Africa do not have access to a healthy diet in which appropriate quantities of all needed nutrients are supplied [10]. Moreover, their diets are characterized by monotony and a lack of food rich in nutrients, which are particularly critical for children [11]. In fact, UNICEF et al. [12] indicated that 6.7% (45.4 million) of children under five years old suffer from wasting, and 22% (149.6 million) of them suffer from stunting globally.

Stunting is a result of malnutrition indicated by a low height-for-age (HAZ) level; wasting, on the other hand, is indicated by a low weight-for-height (WHZ) level; and, lastly, severe acute malnutrition (SAM) is a type of malnutrition defined by the WHO and UNICEF [13] as causing at least one of the following: a low WHZ, a mid-upper arm circumference of < 115 mm, or clinical signs of bilateral oedema.

On the African continent, the number of children under five years old that suffer from wasting and stunting are, respectively, 12.1 million and 64.1 million, with 3.1 million that suffer from SAM. Considering Burundi, the percentages for stunting and wasting in the same category of children are 57.6% and 4.8%; in the Democratic Republic of Congo (DRCongo), stunting affects 40.8% of children, with a wasting percentage of 6.4%. These numbers are particularly worrying if the critical levels (very high) for children in a population are considered: 30% for stunting (40% for alarming) and 15% for wasting (medium: 5–10%, low: 2.5–5%) [12].

To manage situations like those already described, it is appositely thought that high-energy and high-nutrient foods can be a good help. Emergency food products, to be useful, must be ready without preparation requirements when consumed, have a long shelf-life, be easy to distribute everywhere, and, obviously, have high energy and nutrient contents [14]; they must also cover the nutritional needs of different ages, supplying energy content as protein (10–15%), as fat (35–45%), and as carbohydrates (40–50%) [15]. For these reasons, depending on the different situations, ready-to-use nutritional food (RUF) can be divided into ready-to-use therapeutic food (RUTF) and ready-to-use supplementary food (RUSF): the first supplies all of the energy and nutrients necessary for malnourished people’s growth to rapidly catch up; the second supplies only part of a proper diet [16,17]. The classic example of RUTF is Plumpy’Nut, a paste composed of peanut flour, sugar, plant fat, and milk powder and enriched with minerals and vitamins [18]. RUSF products are very diverse, and often they are also made with some local food sources, particularly those rich in protein and fat with relatively lower costs [19]. For example, a bread enriched with different concentrations of oyster mushroom powder (*Pleurotus plumonarius*), a rich source of high-quality proteins and essential amino acids, was developed in Nigeria by Okafor et al. [20]. Agrahar et al. [21] evaluated the enrichment with chickpea flour of the recipe for laddoo, a traditional lemon-sized ball-shaped sweet composed of roasted cereal or legume flour, fat, and sugar, which is very well accepted by people in India and the Middle East. A very common shape to present a therapeutic or supplementary food in is that of a biscuit, since this food shape is easy to be appreciated and consumed by children [22]. High-energy biscuits (HEBs) are very well documented as supplements (because they can easily have protein-rich ingredients and vitamin–mineral premixes added in the formulation) for emergency feeding programs in situations of nutritional deficiencies or a lack of access to basic facilities [23]. The WHO [24] developed the BP-100 biscuit, a milk-based RUTF used by humanitarian agencies and governmental and non-governmental organizations to treat SAM in any cultural setting. Varghese and Srivastav [25] developed a low-cost, high-energy, and nutritious cookie (RUSF) prepared with soy protein isolate, moringa powder, roasted jackfruit seed powder, and a vitamin–mineral premix with the aim of satisfying 30% of the Recommended Dietary Allowance (RDA) requirement for undernourished adolescents in India. A comparison with Plumpy’Nut and two cookies prepared with peanuts, pistachios, maize, and moringa was carried out in Benin for protein and mineral content by Kouton et al. [26]. A study conducted by Fetriyuna et al. [27] evaluated the introduction of peanut into some RUSF biscuits as a protein and lipid source, in conjunction with mung bean, taro flour, and banana nagka.

The objective of this study was to develop an RUSF, using the relatively low-cost food resources that are readily available in Sub-Saharan Africa, with particular focus on Burundi and the Democratic Republic of Congo. In this context, Project C3S (Production of Appropriate Food: Sufficient, Safe and Sustainable), promoted by Università Cattolica del Sacro Cuore (Piacenza), has established pilot centers with the aim of fostering agricultural, technological, and nutritional progress in rural communities of low-income countries.

The initial recipes were developed based on a previous study [28], where three prototypes, with the same objective as this paper, were prepared and characterized. The protein content (12–13%) was deemed insufficiently high, and thus, in the new recipe of the current study, only whole chicken eggs and peanut flour were retained as essential ingredients. The incorporation of raw soy flour and Niebè flour (Vigna unguiculata) was driven by the necessity of enhancing the protein content, while the substitution of sunflower seed oil with palm oil in both recipes and the elimination of wheat flour in the DRCongo recipe were implemented in order to reduce costs. The newly developed biscuits (one for Burundi, and one for the DRCongo) were characterized for their chemical and nutritional composition and tested for stability (with reference to moisture and peroxide content, and texture), simulating typical storage conditions in the target rural context. The new RUSFs are designed to have an impact on children’s nutrition (1–10 years), taking into account their consumption possibilities and the local diets, as well as to be low-cost. The concept has significant potential, contingent upon the implementation of appropriate alterations to the ingredients, in contexts characterized by poverty and rurality where a low-cost RUSF is a necessity for children’s nutritional support, provided that the aforementioned trials in the countries cited above are successful.

## 2. Materials and Methods

### 2.1. Experimental Plan

The development of the RUSF biscuits included the following steps.

Ingredient selection and preparation.Recipe development, with “La Nuova Dieta Ragionata—2003 (3.6.0)” software.Biscuit preparation.Chemical characterization and nutritional evaluation of two final prototypes.Sample production of two final prototypes for shelf-life trial.Monitoring time evolution for moisture, peroxide content, and texture.Evaluation of economic aspects.

### 2.2. Ingredient Selection and Preparation

All the ingredients were sourced from the ethnic market of Piacenza (Italy), with particular attention paid to their availability in Burundi and the DRCongo, particularly in the areas where Project C3S has active projects. Additionally, their costs and traditional varieties and uses were considered. This information was obtained from numerous local informal dietary questionnaires (which do not require ethical approval) within the project [29] (Table 1). Soy flour and Niebè flour (Vigna unguiculata) were prepared by milling the raw grains in “Model 4—Laboratory Mill—Thomas—Wiley” to a particle size of 2 mm.

### 2.3. Recipe Development and Preparation

Using “La Nuova Dieta Ragionata—2003”, a professional software for dieticians, an evaluation was conducted in accordance with Mourey [30] for data pertaining to the nutritional requirements of children. The software database was utilized to assess the chemical characteristics of ingredients, except for Niebè chemical characteristics, which were assessed through laboratory analysis. This evaluation led to the conclusion that a recipe with a protein content of at least 16% and a fat content of the same percentage should be included in the composition of the food. In order to adhere to the aforementioned requirements while maintaining a low price, certain ingredients were substituted, added, or omitted in comparison to the Reggi [28] recipe. Sorghum flour, sweet potato flour, egg yolk, and wheat flour (the latter only present in the DRCongo recipe) were removed. Sunflower seed oil was substituted with palm oil, and raw soy flour and raw Niebè flour were incorporated. In accordance with the aforementioned requirements and utilizing the software, a series of preliminary recipes were devised and subjected to evaluation in order to ascertain the optimal combination of taste and texture. The preparation of prototypes involved the formation of dough through the manual kneading of the dry and wet ingredients for a period of 5 min. Subsequently, the biscuits were shaped (diameter: 3 cm, height: 0.5 cm) and baked in a convection oven (180 °C for 11 min), resulting in biscuits with an approximate weight of 7.1 g each. According to the results of all evaluations, two final biscuits were obtained, one for Burundi and one for the DRCongo (Table 2).

### 2.4. Chemical and Nutritional Characterization

The proximate composition analyses of the two final prototypes were conducted using the following methods: AOAC Official Method 942.05 for ashes, Method ISO 712 for dry matter, AOAC Official Method 920.29 for ether extract (fat), AOAC Official Method 996.11 (using dosage kit K-TSTA, Megazyme International, Bray, Ireland) for starch, AOAC Official Method 2009.01 for fiber, AOAC Official Method 990.03 for proteins, and the method described by Directive 71/250/EEC, 15/6/1971, for reducing sugars [31,32,33]. For the nutritional characterization, the following methods were employed, respectively, for protein digestibility and urease activity: AOAC Official Method 971.09 and Directive 71/250/EEC, 15/6/1971 [31,33]. Once the quantity of biscuits required to fulfill 30% of the protein requirement for children aged between 1 and 10 years had been estimated, the software “La Nuova Dieta Ragionata—2003” was employed, utilizing the requirements suggested by Mourey [30], for a theoretical evaluation of the biscuits’ potential to meet the vitamin A, Fe, and Zn requirements.

### 2.5. Shelf-Life Trial

In order to assess the quality stability of the two biscuits under standard climate conditions, a room-temperature storage test was conducted in Burundi and the DRC using a simple plastic bag packaging. For both recipes, 6 samples of biscuits packed in plastic (PE) lab bags closed with rubber bands were stored in a thermostat at 30 °C in darkness and analyzed at time 0 and after 7, 14, 21, 28, and 35 days. At each time, the biscuits were analyzed for their shelf-life in terms of moisture and peroxide contents, as well as their texture. The moisture content was analyzed using the ISO 712 method [32], following milling with a kitchen mixer. The peroxide content of the fat extracted was evaluated using the method described by Calligaris et al. [34], in accordance with the procedure described in Annex III to Regulation 2568/91/EEC [35]. The peroxide contents are reported as meq O_2_/kg oil. The texture of the biscuits was evaluated according to the method described by Kaur et al. [36], utilizing the Texture Analyzer TVT 6700 with the following settings: height of sample—10 mm, starting distance between sensor and sample—5 mm, starting speed—1 mm/s, test speed—3 mm/s, starting force—50 g, data rate—333 pps. Two parameters were evaluated for texture: hardness (g strength) and fracturability (mm).

### 2.6. Data Analysis

The data for chemical and nutritional parameters (average of triplicate analyses) were compared using the Student test, while the data for stability were compared by ANOVA, with the influence of time and recipe factors also analyzed. Tukey test was employed for post hoc analysis to differentiate the averages. All analyses were conducted using IBM SPSS Statistics (Version 25, SPSS Inc., Chicago, IL, USA). GraphPad Prism 8.0.1 was used for the shelf-life graphs. The results are presented as mean values ± sd, *p* < 0.05.

## 3. Results

### 3.1. Chemical and Nutritional Characterization

The results presented in Table 3 indicate no significant differences between the two recipes with regard to all the parameters. Moreover, it can be observed that a relatively high content of protein (17.81 and 16.77) and fat (19.69 and 20.06) was achieved. The urease activity was found to be very low for both biscuit samples.

The weight of a single biscuit is 7.1 ± 0.1 g, and the mass of biscuits needed to cover 30% of daily protein is shown in Table 4.

Table 5 shows the calculated fulfilment of some micronutrients, according to the daily intake of biscuits.

### 3.2. Economic Considerations

In terms of economic considerations, it is essential to recognize that the analysis presented here has solely focused on the quantity and the retail price/kg of individual ingredients utilized in two recipes, without an evaluation of all the other production costs (machineries, energy, staff, and packaging) and the possible weight loss or processing waste. Subsequently, the prices of the ingredients were collected during the study (April 2021), and the production costs associated with the ingredients purchased were calculated and compared for the two biscuits in both countries. In Burundi, the cost of ingredients for the recipes was found to be 1.471 USD/kg and 1.386 USD/kg, and in the DRCongo 1.312 USD/kg and 1.388 USD/kg, respectively. However, the cost of the two biscuits would be slightly higher in Burundi, with a difference between the two recipes. In Burundi, BUR would be the cheapest (−5.78%), while in the DRCongo it would be the most expensive (+5.48%).

### 3.3. Shelf-Life Trials

With regard to moisture content (Figure 1), the two recipes demonstrated no significant differences over the entire duration of the study, with the same storage times. There were no relevant differences within the same recipe in terms of evolution over time. The sole exception was observed at 21 days for both samples. However, this discrepancy is likely due to a packaging issue with the PE bags used for the 21-day samples.

It can be reasonably assumed that the closure problem also affects the peroxide content (Figure 2), which shows a faster oxidation for BUR compared with CON. In fact, the BUR samples exhibited an increase between 14 and 21 days, while the CON samples demonstrated a similar increase but between 28 and 35 days. Nevertheless, the peroxide values for BUR remained consistently low, suggesting that this trend could be considered almost constant with an oxidation level slightly lower than that of CON.

The texture analysis yielded two distinct parameters: fracturability (Figure 3) and hardness (Figure 4). The fracturability did not significantly vary between the two recipes throughout the entirety of the trial period, aligning with the observed trend in moisture content. In contrast, a significant difference was observed between the two biscuits, with a lower fracturability (higher value in mm run by the sensor before breaking) for BUR than for CON. Concordant trends were observed for the hardness, with constant values over time and higher values for BUR.

## 4. Discussion

The age group most affected by malnutrition, defined as a lack of proper foods (and thus nutrients), is 3–5 years old. During this period, children require more specific dietary components than at other ages, while often, in the Sub-Saharan Africa context, the weaning transition due to slow digestive tract adaptation and the lack of available baby foods can result in inadequate nutrition. Such deficiencies in childhood can have adverse consequences, including an increased incidence of infections; developmental impairment, particularly in the brain and cognitive functions; a heightened risk of poor academic and future occupational performance; and an elevated prevalence of chronic diseases [37]. In an attempt to address the issue of malnutrition, many different therapeutic foods have been developed. One such example is Plumpy’Nut, which is currently being used for child nutrition care in a significant number of developing countries [18]. However, this type of product has certain limitations. Its availability and subsequent distribution are irregular, which hampers a good nutritional recovery process. Furthermore, its characteristics do not align with the typical eating habits of the population, increasing the risk of rejection [38]. One potential solution could be the use of RUSF and RUTF prepared with local available ingredients [26]. This, in fact, can stimulate local economies and increase sustainability, which also has a more beneficial social impact [39], like the possibility of household production and selling preventive food in local markets [40].

In light of the aforementioned considerations, the selected ingredients for the two biscuit recipes are the best compromise between cost effectiveness and nutritional value. The objective is to produce a biscuit that is both rich in nutrients and affordable, with the aim of providing 30% of the daily protein requirement for children. The cost of RUSF and RUTF represents a significant challenge for their availability in low-income countries [41]. For these reasons, milk and other dairy products, and their protein supply, were replaced with the use of egg, soy flour, Niebè flour, and peanut flour. Milk powder represents the most expensive ingredient used in BP-100 [25], a conclusion that was also reached by Sandige et al. [42] and Dibari et al. [43].

Our economic evaluation demonstrates that two distinct recipes are necessary to maintain the lowest possible cost in Burundi and the DRCongo. For instance, the incorporation of wheat flour into the DRCongo recipe would have increased the cost by 5.48%.

Our nutritional evaluation of the biscuits has confirmed the possibility of producing an RUSF suitable to meet 30% of the daily protein requirement with a daily biscuit portion of a maximum of 50 g. This quantity is low enough to be easily consumed and at an affordable cost, allowing the biscuits to be used as a convenient snack between normal meals consumed throughout the day [16,17].

With regard to macronutrients, the BUR and CON biscuits exhibited protein contents of 17.81% and 16.77%, respectively, which were both higher than those of Plumpy’Nut (16.35% [26]) and BP-100 (12.3% [25]), two of the most prevalent and utilized RUFs in low-income countries. This is also true of the biscuits produced by Varghese and Srivastav [25] (13.46%) and Kouton et al. [26] (16.18%; 16.12%). Our driving criterion was the use of low-cost ingredients to enhance the protein content, while also considering the amino acid composition of soy and egg, which is regarded as highly beneficial. The Biological Value (BV) and Protein Digestibility-Corrected Amino Acid Score (PDCAAS) of these two products are, respectively, 100 and 1.00 for egg, and 74 and 1.00 for soy [44].

In terms of fat content, the values for BUR and CON (19.69%; 20.06%) were lower than those observed in BP-100 (24.8%), cookies (25.97%) developed by Varghese and Srivastav [25], and T1 and T2 (respectively, 37.35% and 43.25%) developed by Kouton et al. [26]. In any case, the fat content of our two prototypes is clearly above the 12% of fat defined as standard for high-energy biscuits by USAID [45]. The high lipid values in the recipes are intended to satisfy dietary needs, particularly those pertaining to polyunsaturated fatty acid, given the paucity of such nutrients in routine diets. The relatively low values observed in BUR and CON can be justified by their supportive and non-therapeutic nature.

With regard to the fiber content, the two recipes are distinguished by a remarkably high level of fiber (BUR: 6.63%; CON: 6.95%), which is above the standard maximum defined by USAID [45] for high-energy biscuits (2.3%). This outcome can be attributed to the incorporation of pulse ingredients in the recipes.

With regard to the ash content (BUR: 3.06%; CON: 3.05%), both biscuits exhibited a slightly lower level than the minimum (3.5%) recommended by USAID [45], but this discrepancy can be attributed to the absence of additional minerals not readily available in local markets. The ash content of the two recipes is also lower in comparison with the three biscuits based on mushroom powder (3.51%; 3.89%; 4.23%) of Cornelia and Chandra [14].

The analysis of urease activity revealed values of, respectively, 0.028 (BUR) and 0.02 (CON) mg N/g min at 30 °C, which are ten times lower than the limit defined as 0.23 mg N/g min 30 °C [46]. This evidence demonstrates the capacity of biscuit baking to denature antinutritional factors, such as trypsin inhibitors [47], which are often present in legumes.

The protein digestibility of the biscuits was found to be very high (BUR: 91.72%; CON: 92.01%), which is a significant advantage in terms of suitability for feeding malnourished children.

With regard to the estimated micronutrient content, the two prototypes were not comparable with the similar products presented by Varghese and Srivastav [25] or Fetriyuna et al. [27], who used vitamin and mineral premix in the formulation. It is evident that the two biscuits are unable to fulfil the daily requirements of vitamin A, Zn, and iron for the different age groups. In fact, they can only cover 6–10% of the vitamin A daily need, 13–18% of the Zn daily need, and 25–32% of the Fe daily need. A comparison of the two recipes revealed that CON is the richest one in terms of micronutrients.

With regard to the stability trial, the moisture content of both prototypes remained almost constant, without significant increase, throughout the 35-day period at 30 °C. There were also no significant differences between the two recipes. A comparison with Romani et al. [48] reveals a comparable trend in the temporal evolution of moisture content at similar temperatures, although the data indicate a higher level (3.9–4.1%) in BUR and CON. The humidity in the prototypes also remained below the maximum levels defined by USAID [45] for high-energy biscuits (4.5%), as well as the levels observed in six prototypes created by Beshir et al. [49] with lupin flour (5.32%; 5.50%; 6.38%; 5.75%; 5.44%; 5.37%).

With regard to the peroxide content, the closure problem influenced the trends for the two prototypes, which demonstrated a discernible difference in the rate of lipidic oxidation. This was evidenced by the fact that BUR reached a level higher than 10 Meq O_2_/kg oil at a faster rate than CON, which is considered the starting point for an increase in consumer rejection of similar baking products by Calligaris et al. [34,50]. The latter study [50] defines the rejection area for 70% of the consumers as being between 13.7 and 19.7 Meq O_2_/kg oil, which are levels that were never achieved by the two biscuits during the course of the study. In addition, a comparison with Romani et al. [48] indicates that the maximum peroxide levels in BUR and CON are lower than the maximum levels achieved in similar products tested by these authors (14 Meq O_2_/kg oil).

The final parameter analyzed to assess the product shelf-life was the texture. Two parameters were identified that permit a more detailed explanation of the texture profile: hardness, expressed in g strength, and fracturability, expressed in mm walked by the sensor from minimum to maximum strength (hence, a higher mm indicates a lower fracturability). During the aforementioned period, neither of these two texture parameters exhibited any significant change in either biscuit. However, BUR was found to be harder and, correspondingly, less fracturable than CON. This difference, which does not affect the perception of palatability, can be attributed to the inclusion, in the first prototype, of wheat flour, which represents the sole significant difference between the two formulations. The formation of the gluten network can significantly contribute to the hardness of the biscuit. Similar results were obtained by Lages Rodrigues et al. [51], who conducted a similar texture trial.

## 5. Conclusions

The two biscuit recipes developed for a low-cost ready-to-use supplementary food (RUSF) in the areas followed by Project C3S of Università Cattolica del Sacro Cuore (Piacenza) demonstrate promising results in terms of protein and lipidic integration for the prevention and care of malnutrition in children. The shelf-life test demonstrated that biscuits packaged in a simple PE plastic bag would remain stable for up to 35 days at 30 °C, a temperature compatible with the environmental conditions of Sub-Saharan Africa. The biscuits can be produced with an accessible recipe and with locally available, affordable ingredients, allowing for straightforward distribution, consumption, and further household production in the event that the context does not permit the establishment of industrial facilities.

The subsequent stage of research may entail a feeding trial, with the objective of evaluating the impact of these products on malnourished children.

## Figures and Tables

**Figure 1 foods-13-01614-f001:**
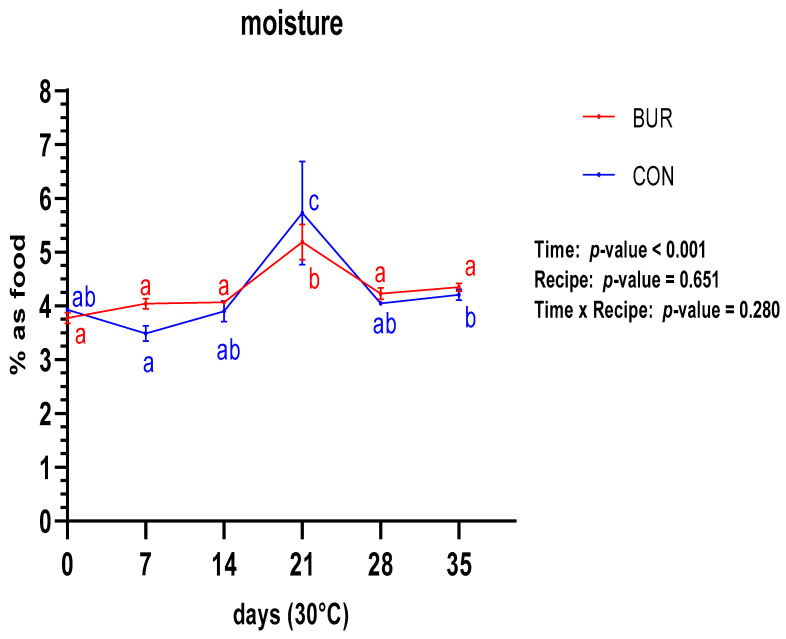
Behavior of moisture content for both recipes during shelf-life trial. a, ab, c—significant differences in the same recipe are indicated with these different letters.

**Figure 2 foods-13-01614-f002:**
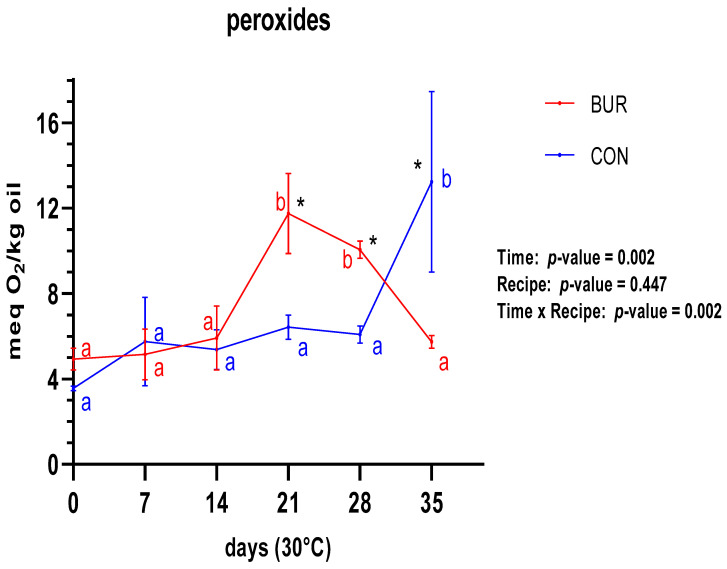
Behavior of peroxide content for both recipes during shelf-life trial. *, a, b—significant differences in the same recipe are indicated with different letters, and significant differences between recipes are indicated with *.

**Figure 3 foods-13-01614-f003:**
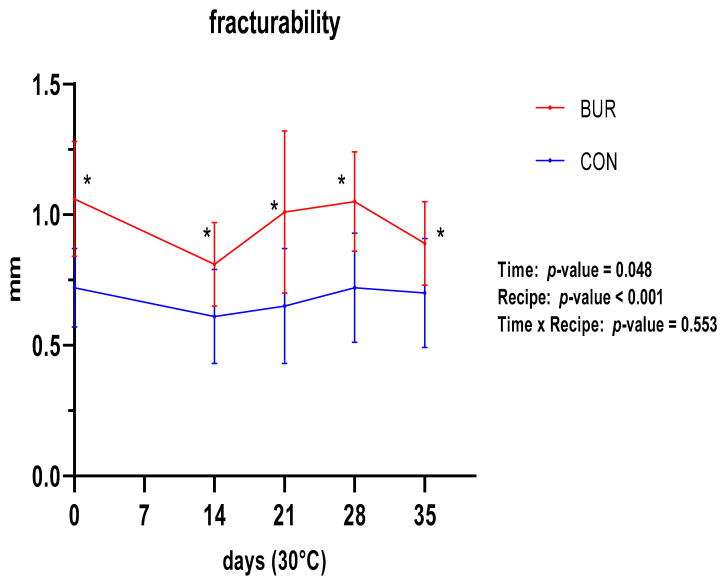
Trends of fracturability for both recipes during shelf-life trial. Significant differences between recipes are indicated with *.

**Figure 4 foods-13-01614-f004:**
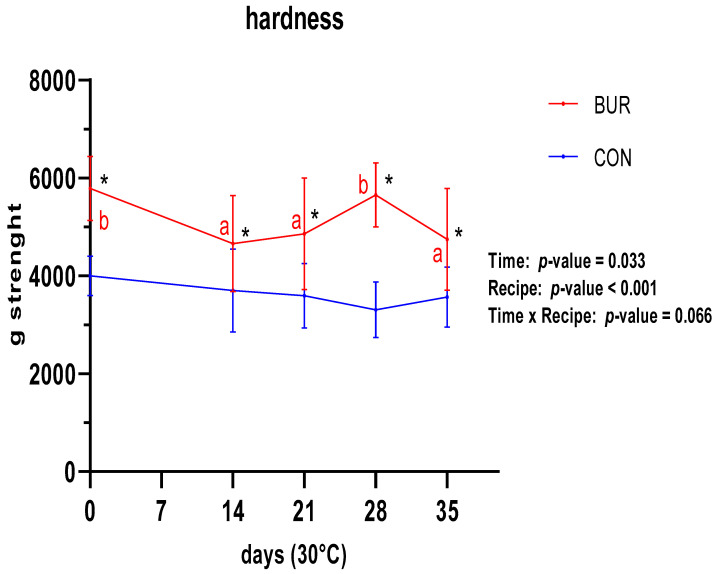
Trends of hardness for both recipes during shelf-life trial. *, a, b—significant differences in the same recipe are indicated with different letters, and significant differences between recipes are indicated with *.

**Table 1 foods-13-01614-t001:** Brand of selected ingredients and their prices (USD/kg) in Burundi and DRCongo (April 2021).

Product	Brand (In Piacenza Market)	Price Burundi (USD/kg)	Price DRCongo (USD/kg)
Whole chicken egg	Terra (Lidl)	2.80	2.93
Raw soy seeds	Calleris	0.73	0.66
Raw Niebè (*Vigna unguiculata*)	Calleris	0.67	0.43
Peanut flour	Kava Foods	1.90	0.58
Rice flour	Adea	0.84	1.08
Wheat flour (type 1)	Cerealpuglia	0.73	2.00
Brown sugar	Tate & Lyle Demerara	0.88	1.40
Palm oil	Blue Bay	0.91	1.00
Baking powder	Paneangeli	1.48	2.11

**Table 2 foods-13-01614-t002:** Composition of biscuits for Burundi and DRCongo (% of wet matter).

Ingredients	BUR (Burundi)	CON (DRCongo)
Whole chicken egg	20	22
Raw soy flour	13	10
Niebè flour	10	15
Peanut flour	16	20
Rice flour	10	10
Wheat flour (type 1)	8	--
Brown sugar	15	15
Palm oil	6	6
Baking powder	2	2

**Table 3 foods-13-01614-t003:** Proximate composition (% as food), protein digestibility (%), and urease (mg N/g min at 30 °C) of the experimental biscuits. Values reported as mean ± sd.

Nutrients	BUR	CON
Dry matter	94.35 ± 0.05	94.17 ± 0.02
Ash	3.06 ± 0.002	3.05 ± 0.004
Protein	17.81 ± 0.03	16.77 ± 0.12
Fat	19.69 ± 0.04	20.06 ± 0.02
Starch	26.37 ± 0.06	22.16 ± 0.01
Fiber	6.63 ± 0.04	6.95 ± 0.02
Soluble fiber	1.56 ± 0.05	1.11 ± 0.02
Insoluble fiber	5.07 ± 0.01	5.84 ± 0.04
Redox sugars	0.19 ± 0.02	0.34 ± 0.03
Protein digestibility	91.72 ± 0.01	92.01 ± 0.03
Urease	0.028 ± 0.003	0.020 ± 0.001

**Table 4 foods-13-01614-t004:** Biscuit weight needed to cover 30% of the daily protein requirement for children 1 to 10 years old (as defined by Mourey [30]).

Children’s Age	Protein Requirement (g/d)	BUR (g/d)	BUR (Number of Biscuits)	CON (g/d)	CON (Number of Biscuits)
1–2 years	4.3	24.14	3.44	25.64	3.61
2–3 years	4.58	25.72	3.62	27.31	3.85
3–4 years	5.39	30.26	4.31	32.14	4.53
4–5 years	5.24	29.42	4.14	31.25	4.4
5–6 years	6.27	35.20	4.96	37.39	5.27
6–7 years	6.21	34.87	4.91	37.03	5.22
7–10 years	8.18	45.93	6.47	48.78	6.87

**Table 5 foods-13-01614-t005:** Calculated fulfilment of micronutrients (vitamin A, Zn, Fe) by the biscuit amounts shown in Table 4 (requirements by Mourey [30]; micronutrients supplied by “La Nuova Dieta Ragionata—2003”).

Children’s Age	Vit. A Needs (μg/Day)	Vit. A Covered (%)		Zn Needs (mg/Day)	Zn Covered (%)		Fe Needs (mg/Day)	Fe Covered (%)	
		BUR	CON		BUR	CON		BUR	CON
1–3 years	400	6.19	7.19	4	16	17.75	3.9	24.61	27.95
4–6 years	400	7.67	8.94	6	13.16	14.66	4.2	28.33	32.14
7–10 years	500	8.51	9.9	7	15.57	17.43	5.9	27.96	31.69

## Data Availability

The original contributions presented in the study are included in the article, further inquiries can be directed to the corresponding author.

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
