# Peer review of "Ready-to-Use Supplementary-Food Biscuit Production with Low-Cost Ingredients for Malnourished Children in Sub-Saharan Africa"

_foods, 2024, doi:10.3390/foods13111614_

Round 1

Reviewer 1 Report

Comments and Suggestions for Authors

I read the paper “LOW-COST RUSF BISCUIT PRODUCTION FOR MALNOURISHED 2 CHILDREN IN SUB-SAHARAN AFRICA” with great interest. Proper methods, well-written in general, only a few minor comments:

1.      Introduction line84, the authors could briefly state the reason why biscuit is easy to be consumed by infants, which I found to be sort of counter-intuitive, especially when RUF in liquid or powder/paste forms also available.

2.      In Table3, what is “DM”? also for testing significance, what is the sample size?

3.      Any palatable test? Which could influence the adherence, as authors also mentioned hardness difference between two biscuits

4.      How did the author think of the low vitamins and minerals of these two biscuits? Did the author consider changing recipes to improve on that later on?

Author Response

Dear reviewer, 

in the following lines the answers to your comments:

1 I changed the sentence, explaining better that the biscuit shape is more simple to be consumed by children considering the usual level of appreciation that children has for this type of product (biscuits and sweets generically)

2 DM is Dry Matter, I modified it in the new manuscript. For the analysis the sample size was three different replicas of the same recipe, and it's explained in paragraph 2.5 - Data analysis

3 Palatable test unfortunately was not possible because we conducted the study in Italy and it was not possible to find a panel suitable to simulate the context of Burundi and DRCongo

4 Our major focus was to improve the protein content, vitamins and minerals are low but it's very difficult to improve their content without using some integration: very expensive in Low - Income Countries like Burundi and DRCongo.

Kind regards,

Davide Reggi

Reviewer 2 Report

Comments and Suggestions for Authors

The topic is remarkable and this work has been soundly written, but in its present form, it is not acceptable and must be revised.

Detailed comments;

The title should not be capitalized.

The abstract should not be extended.

Nearly half of the abstract is related to the importance and objectives of the research. Summarize this section and focus more on your results.

Keywords should not be multi-part phrases.

Results and discussion should be merged.

Line 208, 383, please leave a space between 30 and the Celsius symbol.

One of the antinutritional factors in raw soy flour is a trypsin inhibitor. Did you measure this factor in the prepared biscuits?

Please delete the cost phrase in the title, because you didn't evaluate the production costs.

Change stability to shelf life.

Please add Duncan's letters to the data presented in Table 3 and Figures 3 and 4.

In Table 3, delete the third column.

The title of the figures must be presented under of figures.

Why peroxide value is reduced after 21 days of storage Burundi recipe?

The hardness of Burundi is higher than DRCongo recipe, whereas the protein content is the same! Which factor apart from protein content has affected on hardness of Burundi biscuits?

Regards,

Author Response

Dear reviewers, I followed your suggestions to modify my paper.

Here, in the following lines, some answers to your comments and questions:

I didn't merged the results and discussion chapters because the form suggested by Foods to write research paper asks to separate them

I didn't measure trypsin inhibitor presence because I measured urease activity and the low presence of that is a good indication also for the absence of trypsin inhibitor

For table 3 and figure 3 and 4 I modified the titles explaining better the use of letters and *

Peroxide value is reduced after 21 days in Burundi recipe because the peroxides previously formed were evolved in other compounds and there was a low formation of peroxides in the subsequent 14 days period

The factor that majorly influences the hardness in Burundi recipe is gluten net net formation caused by wheat flour presence. The absence of that ingredient in DRCongo recipe explains the difference of hardness. 

kind regards,

Davide Reggi

Round 2

Reviewer 2 Report

Comments and Suggestions for Authors

ACCEPT